# Faster and Stronger: When ANN-SNN Conversion Meets Parallel Spiking Calculation

Zecheng Hao[1]  Qichao Ma[1]  Kang Chen[1]  Yi Zhang[2]  Zhaofei Yu[1 3]  Tiejun Huang[1 3]

## Abstract

Spiking Neural Network (SNN), as a brain-inspired and energy-efficient network, is currently facing the pivotal challenge of exploring a suitable and efficient learning framework. The predominant training methodologies, namely Spatial-Temporal Back-propagation (STBP) and ANN-SNN Conversion, are encumbered by substantial training overhead or pronounced inference latency, which impedes the advancement of SNNs in scaling to larger networks and navigating intricate application domains. In this work, we propose a novel parallel conversion learning framework, which establishes a mathematical mapping relationship between each time-step of the parallel spiking neurons and the cumulative spike firing rate. We theoretically validate the lossless and sorting properties of the conversion process, as well as pointing out the optimal shifting distance for each step. Furthermore, by integrating the above framework with the distribution-aware error calibration technique, we can achieve efficient conversion towards more general activation functions or training-free circumstance. Extensive experiments have confirmed the significant performance advantages of our method for various conversion cases under ultra-low time latency. To our best knowledge, this is the first work which jointly utilizes parallel spiking calculation and ANN-SNN Conversion, providing a highly promising approach for SNN supervised training. Code is available at https://github.com/hzc1208/Parallel_Conversion.

## 1. Introduction

Spiking Neural Network (SNN), as the third generation of neural networks (Maass, 1997), has become an academic focus in the domain of brain-inspired intelligence. Unlike traditional Artificial Neural Network (ANN), the network backbone of SNN is composed of alternating synaptic layers and neuron layers. Due to the superior biological plasticity and unique firing mechanism of the spiking neuron models, SNNs have great potential in the field of neuromorphic computing and the internal spike triggering events are extremely sparse. At present, SNNs can be effectively deployed on multiple neuromorphic hardwares and have demonstrated significant advantages in inference power consumption (Merolla et al., 2014; Davies et al., 2018; DeBole et al., 2019; Pei et al., 2019).

How to train effective spiking models remains a core topic faced by researchers in the SNN community. The current two mainstream training methods, Spatial-Temporal Back-propagation (STBP) (Wu et al., 2018; Neftci et al., 2019) and ANN-SNN Conversion (Cao et al., 2015; Bu et al., 2022), each have their own advantages and deficiencies, as described in Tab.1. Among them, one can obtain SNN models under ultra-low time latency (*e.g.* $\leq 4 \sim 6$ time-steps) through STBP training, but it requires significant costs in terms of training speed and GPU memory overhead (Yao et al., 2022). Therefore, STBP will struggle greatly for network backbones with larger parameter scale and training time-steps. In addition, when STBP does not fuse global information in the time dimension or adopts fewer time-steps, the performance upper-bound of the trained SNNs still have a gap with that of the pretrained ANNs (Hao et al., 2024b).

The idea of ANN-SNN Conversion is to establish a mathematical mapping relationship between the activation layer of ANNs and the neuron layer of SNNs, so as to replace the activation function modules in pretrained ANNs with corresponding spiking neurons layer by layer, then obtain the converted SNNs. Under this learning framework, the calculation of SNNs only involve the model inference stage, resulting in less training burden and superior performance consistent with pretrained ANNs. But the drawback is that the converted SNN usually requires higher time latency to

[1]School of Computer Science, Peking University, Beijing, China [2]China Mobile Research Institute, Beijing, China [3]Institute for Artificial Intelligence, Peking University, Beijing, China. Correspondence to: Zhaofei Yu <yuzf12@pku.edu.cn>.

*Proceedings of the $42^{nd}$ International Conference on Machine Learning*, Vancouver, Canada. PMLR 267, 2025. Copyright 2025 by the author(s).

Table 1: Comparison of various supervised learning methods for SNNs.

| Method | Act. Func. | Train. Free | Train. Speed | Train. Mem. | Inf. Lat. | Inf. Speed | Inf. Acc. |
|---|---|---|---|---|---|---|---|
| STBP Training | Surro. Func. | ✗ | Slow | Large | Ultra Low | Slow | Low |
| ANN-SNN Conversion | ReLU | ✗ | Fast | Small | Ultra High | Slow | High |
| | QCFS | ✗ | Fast | Small | High | Slow | High |
| Conversion Rect. | QCFS | ✗ | Fast | Small | Low | Slow | High |
| This Work | QCFS | ✗ | Fast | Small | Ultra Low | Fast | High |
| | ReLU | ✓ | N/A | N/A | Low | Fast | High |

reach the ANN-level inference accuracy, especially when using original ReLU ANN directly as the foundation model to achieve the so-called training-free conversion (Li et al., 2021a; Bu et al., 2024). To tackle this problem, researchers have proposed a series of conversion rectification strategies from multiple perspectives, further compressing the inference latency to a small number of time-steps (Wang et al., 2022; Hao et al., 2023b). In addition, due to the fact that the converted SNNs obtained from current conversion schemes are generally based on Integrate-and-Fire (IF) neurons, the inference process of SNNs are limited by serial calculation, which further amplifies the harm of time latency.

Recently, parallel spiking neuron (Fang et al., 2023) is favored by researchers due to its high-speed calculation ability, but current research around it is generally limited to the field of STBP Training. In fact, the training precision of parallel neurons heavily relies on parameter initialization and cannot effectively contribute to the training memory explosion problem of SNNs. Instead, it is likely to play a crucial role in the conversion series methods with higher time latency. In this work, we innovatively combine ANN-SNN Conversion with parallel computing to propose a lossless and high-speed universal parallel conversion framework. The specific contributions are as follows:

- We utilize parallel neurons to map the cumulative spike firing numbers predicted by pretrained ANNs within a specific time period step by step, and theoretically prove the lossless property of the above process.

- We derive the step-wise optimal shifting distance and sorting properties of parallel inference from a mathematical perspective, expanding the applicability of parallel conversion and optimizing its computational overhead.

- We propose a universal learning framework that can achieve effective parallel conversion regardless of (i) the type of activation function used (ii) whether the simulated and actual time-steps are equal.

- Experiments have demonstrated the superior performance of our method for both conventional and training-free conversion. For example, we achieve a

top-1 accuracy of 72.90% on ImageNet-1k, ResNet-34 within merely 4 time-steps.

## 2. Related Works

**STBP training for SNNs.** As a significant learning algorithm that can achieve relatively superior performance for SNNs within ultra-low time latency, Wu et al. (2018) and Neftci et al. (2019) integrated Back-propagation through Time (BPTT) with surrogate gradient to pioneer the concept of STBP. On this basis, researchers have successively combined STBP algorithm with novel BatchNorm (BN) layers (Zheng et al., 2021; Jiang et al., 2024), residual blocks (Fang et al., 2021; Hu et al., 2024), objective learning functions (Li et al., 2021b; Deng et al., 2022; Guo et al., 2022), multi-dimensional attention mechanisms (Qiu et al., 2024), Transformer blocks (Zhou et al., 2023; Shi et al., 2024) and advanced spiking models (Yao et al., 2022; Hao et al., 2024a), thereby extending SNN models to various application scenarios (Ren et al., 2023; Liao et al., 2024; Yao et al., 2024). In addition, to effectively optimize GPU memory overhead and power consumption of STBP training, multiple variant learning frameworks have been further explored, such as time-based learning (Mostafa, 2017) and online learning (Xiao et al., 2022). However, at present, aforementioned schemes still have bottlenecks in learning performance.

**ANN-SNN Conversion.** Compared to STBP training, ANN-SNN Conversion merely needs to replace the activation function modules of the pretrained ANN model with spiking neurons layer by layer to obtain the converted SNN (Cao et al., 2015; Diehl et al., 2015; Sengupta et al., 2019), which not only economizes training load, but also ensures that the converted SNN has sufficiently superior performance upper-bound. Rueckauer et al. (2017) and Han et al. (2020) realized that the soft-reset mechanism is more suitable for conversion learning algorithms, while Deng & Gu (2021) proposed a shiftable ReLU function to better adapt the distribution of the spike firing rate. Based on the above findings, Bu et al. (2022) proposed an advanced quantization activation function and theoretically validated its ability to losslessly simulate the average firing rate within any time latency from the perspective of mathematical expectation.

However, despite achieving better performance than STBP within sufficient time-steps for the same network backbone (Li et al., 2022), converted SNNs still suffer from significant performance degradation under the condition of ultra-low latency (Rathi & Roy, 2021). Therefore, subsequent researchers further proposed more radical error rectification strategies, including setting silent states for specific neurons (Hao et al., 2023a), calibrating forward temporal bias (Wu et al., 2024), shifting initial membrane potential (Hao et al., 2023b), as well as introducing burst and signed spikes (Li & Zeng, 2022; Wang et al., 2022).

**Training-Free Conversion.** How to effectively convert unprocessed ANNs into SNNs with the lowest possible time latency has also become a recent academic focus in the field of conversion learning. Han & Roy (2020) proposed a novel Temporal-Switch-Coding scheme to cut down the time latency and number of addition operations during the SNN inference stage. From the perspectives of calibrating bias and initial membrane potential, Li et al. (2021a) designed two sets of pipelines to further enhance the performance of converted SNNs within dozens of time-steps. Bu et al. (2024) analyzed the upper-bound of the layer-wise conversion error, then pointed out a training-free threshold balancing strategy, which can be applied to various visual tasks on networks with sequential structure.

## 3. Preliminaries

**Spiking Neuron Models.** Leaky Integrate-and-Fire (LIF) neuron is the most commonly-used spiking foundation model in the domain of SNN supervised training. Within a simulation period consisting of $T$ time-steps, $\forall t \in [1, T]$, the LIF neuron will undergo three phases: receiving input current $\mathbf{I}^{l,t}$, firing spikes $\mathbf{s}^{l,t}$, and resetting potential $v_{\text{PRE}}^{l,t}$, which can be described in the following equations:

$$
v_{\text{PRE}}^{l,t} = \lambda^l v^{l,(t-1)} + \mathbf{I}^{l,t}, \ v^{l,t} = v_{\text{PRE}}^{l,t} - \mathbf{s}^{l,t}\theta^l.
$$

$$
\mathbf{I}^{l,t} = \mathbf{W}^l \mathbf{s}^{(l-1),t}\theta^{(l-1)}, \ \mathbf{s}^{l,t} = \begin{cases} 1, & v_{\text{PRE}}^{l,t} \geq \theta^l \\ 0, & \text{otherwise} \end{cases}. \quad (1)
$$

Here $v_{\text{PRE}}^{l,t}$ and $v^{l,t}$ respectively denote the membrane potential before and after firing spikes. $\lambda^l$ and $\theta^l$ regulate the potential leakage degree and firing threshold. IF neuron is a special form of the LIF neuron when $\lambda^l = 1$. $\mathbf{W}^l$ represents the synaptic weight. As the spike firing process of the LIF neuron depends on the value of the previous membrane potential, the calculation procedure in each layer is serial, which limits the inference speed of SNNs. To address this issue, Fang et al. (2023) proposed the concept of parallel

spike computing:

$$
v_{\text{PRE}}^l = \mathbf{\Lambda}^l \mathbf{I}^l, \ \mathbf{\Lambda}^l = \begin{bmatrix} 1 & 0 & \cdots & 0 \\ \lambda^l & 1 & \cdots & 0 \\ \vdots & \vdots & \ddots & \vdots \\ (\lambda^l)^{T-1} & (\lambda^l)^{T-2} & \cdots & 1 \end{bmatrix}.
$$

$$(2)$$

Here $\mathbf{\Lambda}^l \in \mathbb{R}^{T \times T}$. As the dynamic equation of the LIF neuron at the $t$-th time-step can also be rewritten as $v_{\text{PRE}}^{l,t} = \sum_{i=1}^{t}(\lambda^l)^{t-i}\mathbf{I}^{l,i} - \sum_{i=1}^{t-1}(\lambda^l)^{t-i}\mathbf{s}^{l,i}$, when we ignore the influence of the previous spike firing sequence $[\mathbf{s}^{l,1}, ..., \mathbf{s}^{l,(t-1)}]$ on the current time-step, the equation will degenerate into the form of Eq.(2), which can finish the calculation process of $T$ time-steps in parallel at once. However, when $\lambda^l$ is not a small value (e.g. $\lambda^l = 1$), the contribution of the previous spike sequence to the current time-step (i.e. $-\sum_{i=1}^{t-1}(\lambda^l)^{t-i}\mathbf{s}^{l,i}$) cannot be directly ignored, causing the calculation result of the above parallel scheme to deviate from that of vanilla LIF neuron. In addition, when $\mathbf{\Lambda}^l$ is set as learnable parameters (e.g. in STBP training), some potential inappropriate values (e.g. $\forall i, j \in [1, T], \mathbf{\Lambda}_{ij}^l < 0$) may lead to the problem of gradient vanishing.

**ANN-SNN Conversion.** Vanilla conversion methods are generally based on the approximate linear transformation relationship of the average spike firing rates from pre-synaptic and post-synaptic layers. Specifically, when we combine the charging and resetting process as mentioned in Eq.(1), we will have:

$$
\mathbf{s}^{l,t}\theta^l = \mathbf{W}^l \mathbf{s}^{(l-1),t}\theta^{(l-1)} - \left(v^{l,t} - \lambda^l v^{l,(t-1)}\right). \quad (3)
$$

Then, if we consider IF neuron and calculate the average firing situation along time dimension for both sides of the equation, we can further obtain:

$$
\mathbf{r}^{l,T} = \mathbf{W}^l \mathbf{r}^{(l-1),T} - \frac{v^{l,T} - v^{l,0}}{T}. \quad (4)
$$

Here we use $\mathbf{r}^{l,T} = \sum_{t=1}^{T} \mathbf{s}^{l,t}\theta^l / T$ to denote the average firing rate. Bu et al. (2022) further pointed out that when $v^{l,0} = \theta^l/2$ and assuming that the spike sequence is uniformly distributed in time dimension, we can utilize the following Quantization-Clip-Floor-Shift (QCFS) function to simulate the average spike firing rate within any number of time-steps:

$$
\mathbf{r}_{\text{QCFS}}^{l,\tilde{T}} = \frac{\theta^l}{\tilde{T}} \text{Clip}\left(\left\lfloor \frac{\mathbf{W}^l \mathbf{r}^{(l-1),\tilde{T}}\tilde{T} + \psi^l}{\theta^l} \right\rfloor, 0, \tilde{T}\right). \quad (5)
$$

Here $\tilde{T}$ and $\psi^l$ respectively represents the simulation time period and shift term in QCFS function. Whether $\tilde{T}$ is equal to $T$ or not, when $\psi^l = \theta^l/2$, $\mathbf{r}_{\text{QCFS}}^{l,\tilde{T}}$ and $\mathbf{r}^{l,T}$ always maintain equivalence from the perspective of mathematical expectation.

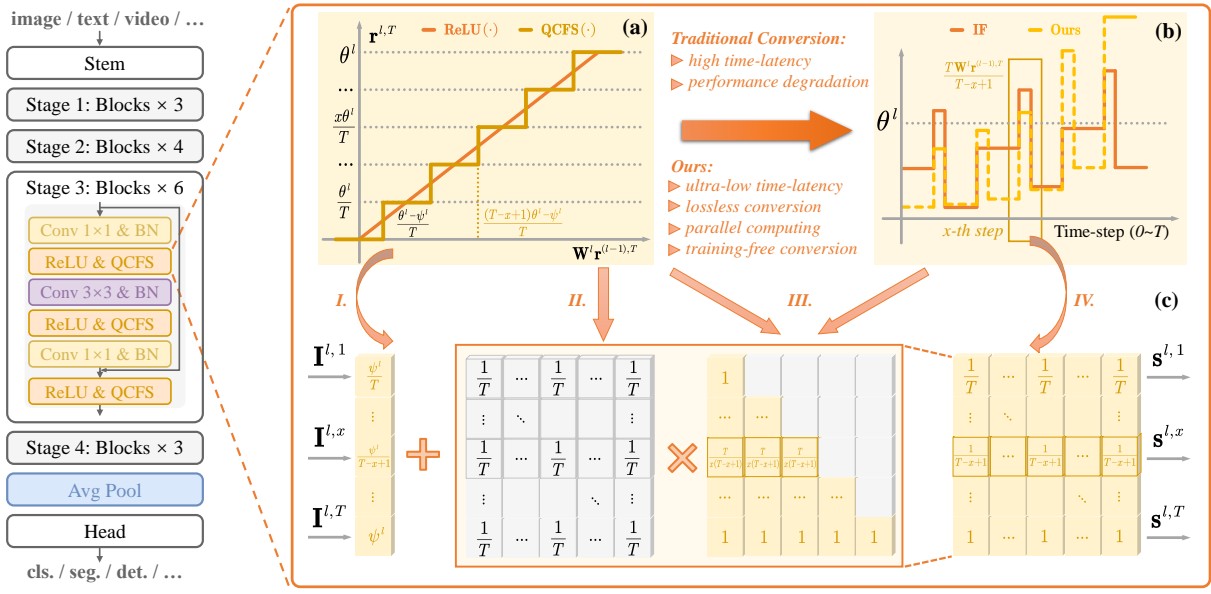

Figure 1: The overall framework of parallel conversion. Here (a) depicts the activation functions in ANNs, (b) shows the sorting property of parallel spiking neurons in the firing phase, and (c) describes the specific process of parallel inference.

## 4. Methods

### 4.1. Establishing an Equivalent Mapping Relationship between Spiking Parallel Inference and QCFS

For traditional conversion framework, spiking neurons generally require a higher time latency (*i.e.* larger $T$) to make the error term $(\boldsymbol{v}^{l,T} - \boldsymbol{v}^{l,0})/T$ in Eq.(4) approach zero, thereby achieving the layer-wise output alignment between the pretrained ANN and converted SNN. Considering the inherent serial computing property of LIF neurons, the total time cost of SNN inference stage will be further exacerbated. Unlike previous approaches, here we attempt to explore the mathematical mapping relationship between parallel spiking neurons and ANN activation functions. Intuitively, a larger average input current $\mathbf{W}^l\mathbf{r}^{(l-1),T}$ will cause spiking neurons to emit spikes earlier. Therefore, for a QCFS function with $T$ quantization levels, at the $x$-th time-step ($x \in [1, T]$), we will judge whether the total number of firing spikes is not less than $T - x + 1$, as shown in Fig.1-*III,IV*.

Specifically, we first define a primary parallel conversion matrix by refering to Eq.(2):

$$\boldsymbol{\Lambda}_{\text{POST}}^l = \begin{bmatrix} \mathbf{c}^{l,1} \\ \mathbf{c}^{l,2} \\ \vdots \\ \mathbf{c}^{l,T} \end{bmatrix} \odot \begin{bmatrix} 1 & 0 & \cdots & 0 \\ 1 & 1 & \cdots & 0 \\ \vdots & \vdots & \ddots & \vdots \\ 1 & 1 & \cdots & 1 \end{bmatrix}. \quad (6)$$

To simplify the calculation, we assume that the ratio of the input current obtained at the $x$-th time-step is $\mathbf{c}^{l,x}$. Since we need to determine whether $\mathbf{W}^l\mathbf{r}^{(l-1),T}T \geq (T - x + 1)\theta^l$

or not at this time-step, we can derive:

$$\begin{cases} x \cdot \mathbf{c}^{l,x} \cdot \mathbf{W}^l\mathbf{r}^{(l-1),T} = \theta^l \\ \mathbf{W}^l\mathbf{r}^{(l-1),T}T = (T-x+1)\theta^l \end{cases} \Rightarrow \mathbf{c}^{l,x} = \frac{T}{x(T-x+1)}. \quad (7)$$

Eq.(7) indicates that when $\mathbf{W}^l\mathbf{r}^{(l-1),T}T \geq (T - x + 1)\theta^l$, we will have $\boldsymbol{\Lambda}_{\text{POST}}^{l,x}\mathbf{I}^l \geq \theta^l$, then neurons will emit spikes to the posterior layer; Otherwise, spiking neurons will remain silent. Therefore, for $(T - x + 2)\theta^l > \mathbf{W}^l\mathbf{r}^{(l-1),T}T \geq (T - x + 1)\theta^l$, spiking neurons will continuously fire spikes from the $x$-th step to the $T$-th step, with a total firing count of $T - x + 1$, which is completely consistent with the simulated result of the corresponding QCFS ANN.

However, it is worth noting that the calculation in Eq.(7) is based on the assumption that the input current completely follows a uniform distribution (*i.e.* $\forall x \in [1, T], \mathbf{I}^{l,x} = \mathbf{W}^l\mathbf{r}^{(l-1),T}$). From the above analysis, one can find that the spike sequence transmitted from the $l$-th layer to the $l + 1$-th layer $[\mathbf{s}^{l,1}, ..., \mathbf{s}^{l,T}]$ clearly does not satisfy the condition of uniform distribution. To effectively regulate the distribution of the input current layer by layer, we introduce the concept of conversion premise controling matrix: $\boldsymbol{\Lambda}_{\text{PRE}}^l = \frac{1}{T} \cdot \mathbf{1}, \mathbf{1} \in \mathbb{R}^{T \times T}$, as shown in Fig.1-*II*.

For each layer in the converted SNN, the input current is first projected into a uniformly distributed state through $\boldsymbol{\Lambda}_{\text{PRE}}^l$, and then transformed into a spike sequence consistent with the ANN prediction through $\boldsymbol{\Lambda}_{\text{POST}}^l$ and parallel spiking neuron. We perform re-parameterization fusion for $\boldsymbol{\Lambda}_{\text{PRE}}^l$ and $\boldsymbol{\Lambda}_{\text{POST}}^l$ to obtain the final parallel conversion matrix $\boldsymbol{\Lambda}_{\text{PC}}^l$:

$$\Lambda_{\text{PC}}^l = \Lambda_{\text{POST}}^l \Lambda_{\text{PRE}}^l = \begin{bmatrix} \frac{1}{T} & \frac{1}{T} & \cdots & \frac{1}{T} \\ \frac{1}{T-1} & \frac{1}{T-1} & \cdots & \frac{1}{T-1} \\ \vdots & \vdots & \ddots & \vdots \\ 1 & 1 & \cdots & 1 \end{bmatrix}. \quad (8)$$

Next, we will consider the transformation towards the shift term in QCFS function (Fig.1-$I$). For multi-step parallel inference, $\psi^l$ plays a role similar to the initial membrane potential, but the contribution of $\psi^l$ to different time-steps is obviously various. We theoretically prove the optimal value of the shift term, which can achieve lossless conversion from QCFS function to SNN parallel inference:

**Theorem 4.1.** *For a $T$-steps parallel inference in the $l$-th layer, we use $\mathbf{b}^l$ to denote the corresponding shift term, here $\mathbf{b}^l \in \mathbb{R}^T$. When the pretrained ANN adopts QCFS function in Eq.(5), for the following cases, we will derive the optimal value of the shift term: $\mathbf{b}^l = \left[ \frac{\psi^l}{T} \cdots \frac{\psi^l}{T-x+1} \cdots \psi^l \right]^\top$.*
*(i) If $T = \tilde{T}$, then we have $\mathbf{r}^{l,T} = \mathbf{r}_{\text{QCFS}}^{l,\tilde{T}}$.*
*(ii) If $T \neq \tilde{T}$ and $\psi^l = \theta^l/2$, then we have $\mathbb{E}\left( \mathbf{r}^{l,T} - \mathbf{r}_{\text{QCFS}}^{l,\tilde{T}} \right) = \mathbf{0}$.*

### 4.2. Towards Universal Conversion Error Rectification

Theorem 4.1(ii) indicates that under the condition of receiving uniform data distribution, even if the simulated time latency $\tilde{T}$ is not equal to the actual inference latency $T$, from the perspective of mathematical expectation, the conversion error can still be cut down to zero. However, on the one hand, the input data may not necessarily follow a uniform distribution in reality; on the other hand, there may be significant distribution gap across different channels. Therefore, to further rectify the conversion error under arbitrary data distribution and time latency, we propose a Distribution-Aware QCFS (DA-QCFS) function:

$$\mathbf{r}_{\text{DA}}^{l,\tilde{T}} = \frac{\theta^l + \phi_{\text{DA}}^l}{\tilde{T}} \text{Clip}\left( \left\lfloor \frac{(\mathbf{W}^l \mathbf{r}^{(l-1),\tilde{T}} + \psi_{\text{DA}}^l)\tilde{T} + \psi^l}{\theta^l} \right\rfloor, 0, \tilde{T} \right). \quad (9)$$

Here $\psi_{\text{DA}}^l, \phi_{\text{DA}}^l \in \mathbb{R}^C$, which are the learnable shifting and scaling factors of DA-QCFS function ($C$ denotes the total number of channels). For each layer, along the channel dimension, we respectively calculate the mean conversion errors $\mathbf{e}_{\text{PRE}}^l, \mathbf{e}_{\text{POST}}^l$ before and after the activation function, which are then used to update $\phi_{\text{DA}}^l, \psi_{\text{DA}}^l$ iteratively. The overall learning process of $\phi_{\text{DA}}^l, \psi_{\text{DA}}^l$ adopts the idea of greedy algorithm. The goal of each iterative update is to make the distribution of the average firing rate during actual inference closer to the distribution simulated by the pretrained ANN, thereby reducing the precision loss when converting from the original model.

Subsequently, we can also losslessly convert DA-QCFS function into parallel spiking neurons. The specific process has been described in Algorithm 1.

---

**Algorithm 1** The overall pseudo-code for universal parallel conversion

---

**Require:** Pretrained ANN model $f_{\text{ANN}}$ with $L$ layers; original activation function (QCFS or ClipReLU) $\boldsymbol{g}_{\text{OA}}^l(\cdot)$ and layer-wise output $\mathbf{r}_{\text{OA}}^l$; actual activation function (DA-QCFS) $\boldsymbol{g}_{\text{DA}}^l(\cdot)$ and layer-wise output $\mathbf{r}_{\text{DA}}^{l,T}$; calibration dataset $D$; mean function along the channel dimension $\mu(\cdot)$; learning momentum $\alpha$.
**Ensure:** Converted Parallel SNN model $f_{\text{SNN}}$.
  **# Stage I: Parameter Initialization**
  **for** $l = 1$ to $L$ **do**
    $f_{\text{ANN}}.\boldsymbol{g}_{\text{DA}}^l.\theta^l = f_{\text{ANN}}.\boldsymbol{g}_{\text{OA}}^l.\theta^l$
    $f_{\text{ANN}}.\psi_{\text{DA}}^l = \mathbf{0}$
    $f_{\text{ANN}}.\phi_{\text{DA}}^l = \mathbf{0}$
  **end for**
  **# Stage II: Layer-wise Error Calibration**
  **for** (**Image**, **Label**) in $D$ **do**
    **for** $l = 1$ to $L$ **do**
      $\mathbf{e}_{\text{PRE}}^l = \mu\left( \mathbf{W}^l \mathbf{r}_{\text{OA}}^{(l-1)} - \mathbf{W}^l \mathbf{r}_{\text{DA}}^{(l-1),T} \right)$
      $f_{\text{ANN}}.\psi_{\text{DA}}^l = \alpha \cdot f_{\text{ANN}}.\psi_{\text{DA}}^l + (1-\alpha) \cdot \mathbf{e}_{\text{PRE}}^l$
      Refering to Eq.(5) and Eq.(9), calculate the original and actual activation output $\mathbf{r}_{\text{OA}}^l, \mathbf{r}_{\text{DA}}^{l,T}$ through $f_{\text{ANN}}.\boldsymbol{g}_{\text{OA}}^l(\cdot)$ and $f_{\text{ANN}}.\boldsymbol{g}_{\text{DA}}^l(\cdot)$
      $\mathbf{e}_{\text{POST}}^l = \mu\left( \mathbf{r}_{\text{OA}}^l - \mathbf{r}_{\text{DA}}^{l,T} \right)$
      $f_{\text{ANN}}.\phi_{\text{DA}}^l = \alpha \cdot f_{\text{ANN}}.\phi_{\text{DA}}^l + (1-\alpha) \cdot \mathbf{e}_{\text{POST}}^l$
    **end for**
  **end for**
  **# Stage III: Parallel Conversion**
  **for** $l = 1$ to $L$ **do**
    **for** $t = 1$ to $T$ **do**
      $f_{\text{SNN}}.\mathbf{b}^{l,t} = \frac{f_{\text{ANN}}.\boldsymbol{g}_{\text{DA}}^l.\theta^l}{2(T-t+1)} + \frac{f_{\text{ANN}}.\psi_{\text{DA}}^l \cdot T}{T-t+1}$
    **end for**
    $f_{\text{SNN}}.\mathbf{W}^l = f_{\text{ANN}}.\mathbf{W}^l$
    Set $f_{\text{SNN}}.\Lambda_{\text{PC}}^l$ according to Eq.(8)
    $f_{\text{SNN}}.\theta_{\text{PRE}}^l = f_{\text{ANN}}.\boldsymbol{g}_{\text{DA}}^l.\theta^l$
    $f_{\text{SNN}}.\theta_{\text{POST}}^l = f_{\text{ANN}}.\boldsymbol{g}_{\text{DA}}^l.\theta^l + f_{\text{ANN}}.\phi_{\text{DA}}^l$
  **end for**
  **# Stage IV: Parallel Inference**
  **for** $l = 1$ to $L, t = 1$ to $T$ **do**
    $\mathbf{I}^l = f_{\text{SNN}}.\mathbf{W}^l \mathbf{s}^{(l-1)} f_{\text{SNN}}.\theta_{\text{POST}}^{(l-1)}$
    **if** $f_{\text{SNN}}.\Lambda_{\text{PC}}^l \mathbf{I}^l + f_{\text{SNN}}.\mathbf{b}^l \geq f_{\text{SNN}}.\theta_{\text{PRE}}^l$ **then**
      Fire spikes: $\mathbf{s}^l = \mathbf{1}$
    **else**
      Keep slient: $\mathbf{s}^l = \mathbf{0}$
    **end if**
  **end for**
  **Return** $f_{\text{SNN}}(\mathbf{W}, \Lambda_{\text{PC}}, \mathbf{b}, \theta_{\text{PRE}}, \theta_{\text{POST}})$.

---

It is worth noting that the above algorithm is also applicable

Table 2: Detailed experimental configuration for universal parallel conversion framework.

| Conversion Cases | Need Thre. Rec. | Need Calib. | $\Lambda_{\mathrm{PC}}^{l}$.shape | $\mathbf{b}^l$.shape | $\theta_{\mathrm{PRE}}^l$.shape | $\theta_{\mathrm{POST}}^l$.shape |
|---|---|---|---|---|---|---|
| QCFS ($\tilde{T} = T$) | ✗ | ✗ | $[T, T]$ | $[T,]$ | scalar | scalar |
| QCFS ($\tilde{T} \neq T$) | ✗ | ✓ | $[T, T]$ | $[T, C]$ | scalar | $[C,]$ |
| ReLU | ✓ | ✓ | $[T, T]$ | $[T, C]$ | $[C,]$ | $[C,]$ |

to the training-free conversion for pretrained ReLU ANNs. Compared to QCFS ANN family, which is specifically designed for conversion learning, a large number of models in the ANN community are typically based on vanilla ReLU function. The data distribution based on ReLU is more irregular and has a larger numerical range, which is more difficult to achieve precise conversion under low time latency. Here we propose a three-stage training free conversion framework:

- **From ReLU to ClipReLU.** For each layer, we utilize the calibration dataset to record the historical maximum activation value within each channel, then set it as $\theta^l$ to achieve the transformation from $\mathrm{ReLU}(\cdot) = \max(0, \cdot)$ to $\mathrm{ClipReLU}(\cdot, 0, \theta^l) = \min\left(\max(0, \cdot), \theta^l\right)$.

- **From ClipReLU to DA-QCFS.** As shown in Algorithm 1, for the specified inference time $T$, we will replace ClipReLU (set as $g_{\mathrm{OA}}^l$) with $T$-level initialized DA-QCFS (set as $g_{\mathrm{DA}}^l$). Due to the fact that the actual data distribution may be irregular, then we adopt layer-wise error calibration to enhance the inference performance within $T$ time-steps as much as possible.

- **From DA-QCFS to parallel spiking neuron.** Specifically, as illustrated in Algorithm 1, $\psi_{\mathrm{DA}}^l$ and $\psi^l$ can be merged together in the bias term; for $\phi_{\mathrm{DA}}^l$, we can achieve mathematical equivalent mapping by setting pre-threshold $\theta_{\mathrm{PRE}}^l$ and post-threshold $\theta_{\mathrm{POST}}^l$.

Overall, we utilize a unified foundational framework for pretrained ANNs based on QCFS ($\tilde{T} = T, \tilde{T} \neq T$) or ReLU, with the only difference being whether additional threshold recording (ReLU $\rightarrow$ ClipReLU) and error calibration stages (ClipReLU/QCFS $\rightarrow$ DA-QCFS) are introduced. Among them, threshold recording has no accuracy loss on the calibration dataset, error calibration aims to reduce the error to zero from the level of mathematical expectation, while parallel conversion is completely lossless for any data distribution. The detailed comparison of the above three conversion cases has been listed in Tab.2.

### 4.3. Optimizing the Calculation Overhead of Spiking Parallel Inference

In the previous discussion, we have pointed out that for the parallel conversion matrix in Eq.(8), the calculation intention of $\Lambda_{\mathrm{PC}}^{l,x}$ is to determine whether the total number of firing spikes within $T$ time-steps is not less than $T - x + 1$. That is to say, if parallel neurons emit spikes at the $x$-th step, they will continue to emit spikes from the $x+1$-th step to the $T$-th step. Therefore, to further optimize the computational overhead and inference speed, we can leverage this sorting property and apply the binary search technique in the parallel inference stage.

Specifically, under the initial state, we respectively set lower-bound and upper-bound pointers $\mathbf{ptr}_{\mathrm{L}}^l, \mathbf{ptr}_{\mathrm{U}}^l$ for the search interval, where $\mathbf{ptr}_{\mathrm{L}}^l = 1, \mathbf{ptr}_{\mathrm{U}}^l = T$. In each subsequent search, we select the $\mathbf{mid} = \left\lfloor \frac{\mathbf{ptr}_{\mathrm{L}}^l + \mathbf{ptr}_{\mathrm{U}}^l}{2} \right\rfloor$-th step and calculate $\Lambda_{\mathrm{PC}}^{l,\mathbf{mid}} \mathbf{I}^l + \mathbf{b}^{l,\mathbf{mid}}$. If $\mathbf{s}^{l,\mathbf{mid}} = \mathbf{1}$, the next search interval will be squeezed to $[\mathbf{ptr}_{\mathrm{L}}^l, \mathbf{mid}]$, otherwise it will be updated to $[\mathbf{mid} + 1, \mathbf{ptr}_{\mathrm{U}}^l]$. Finally, we will derive the time-step $\mathbf{t}_{\mathrm{FIR}}$ at which the first spike is emitted. Then, we can directly set $\mathbf{s}^{l,\mathbf{t}_{\mathrm{FIR}}:T} = \mathbf{1}, \mathbf{s}^{l,1:\mathbf{t}_{\mathrm{FIR}}-1} = \mathbf{0}$ and transmit it to the next synaptic layer.

In addition, during the actual inference process, we can choose $\left[1, ..., \frac{T}{T-x+1}, ..., T\right]^\top \odot \mathbf{W}^l \mathbf{r}^{(l-1),T}$ rather than $\Lambda_{\mathrm{PC}}^l \mathbf{I}^l$. Obviously, the above schemes are computationally equivalent, but the Hadamard product can further reduce the total number of charging operations from $O(T^2)$ to $O(T)$.

Overall, by combining the above two optimization techniques, compared to vanilla LIF neuron, we can achieve the calculation of charging phase within $O(T)$ and derive the complete spike firing sequence within only $O(\log T)$, without the need for additional reset phase.

## 5. Experiments

Consistent with previous conversion learning works, we conduct performance validation on CIFAR (Krizhevsky et al., 2009) and ImageNet (Deng et al., 2009) datasets by using two types of network backbones, VGG (Simonyan & Zisserman, 2014) and ResNet (He et al., 2016). We selected multiple methods including STBP Training (Li et al., 2021b, Dspike; Guo et al., 2022, RecDis; Yao et al., 2022, GLIF;

Table 3: Comparison of previous state-of-the-art learning methods. † denotes adopting the error calibration technique.

| Dataset | Method | Type | ANN Acc.(%) | Arch. | T | SNN Acc.(%) |
|---------|--------|------|-------------|-------|---|-------------|
| CIFAR-10 | OPT | ANN-SNN Conversion | 93.51 | VGG-16 | 32 | 88.79 |
| | QCFS | ANN-SNN Conversion | 95.52 | VGG-16 | 2, 4, 8 | 91.18, 93.96, 94.95 |
| | SNM | Conversion Rect. | 94.09 | VGG-16 | 32 | 93.43 |
| | SRP | Conversion Rect. | 95.52 | VGG-16 | 6 (4+2) | 94.47 |
| | **Ours** | **Parallel Conversion** | 95.43 | VGG-16 | 2, 4 | **94.16, 95.50** |
| | QCFS | ANN-SNN Conversion | 91.77 | ResNet-20 | 2, 4, 8 | 73.20, 83.75, 89.55 |
| | SRP | Conversion Rect. | 91.77 | ResNet-20 | 6 (4+2) | 88.73 |
| | **Ours** | **Parallel Conversion** | 91.67 | ResNet-20 | 2, 4 | **87.42, 91.58** |
| CIFAR-100 | OPT | ANN-SNN Conversion | 70.21 | VGG-16 | 32 | 56.16 |
| | QCFS | ANN-SNN Conversion | 76.28 | VGG-16 | 2, 4, 8 | 63.79, 69.62, 73.96 |
| | SNM | Conversion Rect. | 74.13 | VGG-16 | 32 | 71.80 |
| | SRP | Conversion Rect. | 76.28 | VGG-16 | 6 (4+2) | 74.31 |
| | **Ours** | **Parallel Conversion** | 76.11 | VGG-16 | 2, 4 | **72.71, 75.98** |
| | QCFS | ANN-SNN Conversion | 69.94 | ResNet-20 | 4, 8, 16 | 34.14, 55.37, 67.33 |
| | SRP | Conversion Rect. | 69.94 | ResNet-20 | 6 (4+2) | 53.96 |
| | **Ours** | **Parallel Conversion** | 69.57 | ResNet-20 | 4, 8 | **65.31, 69.62** |
| ImageNet-1k | OPT | ANN-SNN Conversion | 72.40 | VGG-16 | 32 | 54.92 |
| | QCFS | ANN-SNN Conversion | 74.29 | VGG-16 | 8, 16, 32 | 19.12, 50.97, 68.47 |
| | SNM | Conversion Rect. | 73.18 | VGG-16 | 32 | 64.78 |
| | Burst | Conversion Rect. | 74.27 | VGG-16 | 32 | 70.61 |
| | COS | Conversion Rect. | 74.19 | VGG-16 | 10 (8+2) | 70.59 |
| | **Ours** | **Parallel Conversion** | 74.23 | VGG-16 | 4, 8, 16 | **71.23, 73.92, 74.26** |
| | **Ours**† | **Parallel Conversion** | 74.23 | VGG-16 | 4 | **71.75** |
| | RecDis | STBP Training | - | ResNet-34 | 6 | 67.33 |
| | Dspike | STBP Training | - | ResNet-34 | 6 | 68.19 |
| | GLIF | STBP Training | - | ResNet-34 | 4 | 67.52 |
| | TAB | STBP Training | - | ResNet-34 | 4 | 67.78 |
| | OPT | ANN-SNN Conversion | 70.95 | ResNet-34 | 64 | 59.52 |
| | QCFS | ANN-SNN Conversion | 74.32 | ResNet-34 | 8, 16, 32 | 35.06, 59.35, 69.37 |
| | COS | Conversion Rect. | 74.22 | ResNet-34 | 10 (8+2) | 72.66 |
| | **Ours** | **Parallel Conversion** | 74.30 | ResNet-34 | 4, 8 | **67.28, 74.32** |
| | **Ours**† | **Parallel Conversion** | 74.30 | ResNet-34 | 4 | **72.90** |

Jiang et al., 2024, TAB), ANN-SNN Conversion (Deng & Gu, 2021, OPT; Bu et al., 2022, QCFS), Conversion Rectification (Li & Zeng, 2022, Burst; Wang et al., 2022, SNM; Hao et al., 2023a, SRP; Hao et al., 2023b, COS), and Training-Free Conversion (Li et al., 2021a, SNNC; Bu et al., 2024, TBC) as comparison targets. The detailed experimental configuration is provided in Appendix.

### 5.1. Comparison with Previous state-of-the-art Works

In Tab.3, we choose QCFS ANNs as the pretrained base models, and the hyper-parameter settings of QCFS function are the same as (Bu et al., 2022) ($\tilde{T} = 8$ for CIFAR-100/ImageNet-1k, ResNet-20/34; $\tilde{T} = 16$ for ImagNet-1k, VGG-16; $\tilde{T} = 4$ for the remaining cases). One can note that when the inference latency $T$ is equal to the simulation latency $\tilde{T}$, the performance of converted SNNs is generally at the same level as that of the corresponding ANNs. When

$T \ll \tilde{T}$, especially for complex datasets and deeper network backbones, the additional utilization of layer-wise error calibration will further enhance the performance of SNNs under ultra-low latency.

Compared with other types of learning methods, our approach has achieved significant advantages, even surpassing the memory-hungry STBP methods, which means that the parallel conversion scheme may open up a third path for the domain of SNN supervised learning besides STBP and ANN-SNN Conversion. For instance, we achieve 73.92% for ImageNet-1k, VGG-16 within 8 time-steps, which exceeds the performance of COS ($T = 10$) by 3.33% and is at least 3.31% higher than the reported accuracies of remaining methods even if extending the inference latency by $4\times$ (*i.e.* $T = 32$).

Table 4: Comparison of training-free conversion algorithms on ImageNet-1k dataset. † denotes utilizing fine-tuning training.

| Method | Arch. | ANN Acc.(%) | $T = 8$ | $T = 16$ | $T = 32$ | $T = 64$ |
|---|---|---|---|---|---|---|
| TBC | ResNet-18 | 69.76 | - | - | 50.65 (-19.11) | 64.79 (-4.97) |
| SNNC-LP | ResNet-34 | 75.66 | - | - | 50.21 (-25.45) | 63.66 (-12.00) |
| SNNC-AP† | ResNet-34 | 75.66 | - | - | 64.54 (-11.12) | 71.12 (-4.54) |
| TBC | ResNet-34 | 73.31 | - | - | 59.03 (-14.28) | 70.47 (-2.84) |
| | ResNet-18 | 69.76 | 55.18 (-14.58) | 66.26 (-3.50) | 69.05 (-0.71) | 69.54 (-0.22) |
| **Ours** | ResNet-34 | 73.31 | 50.67 (-22.64) | 68.04 (-5.27) | 72.46 (-0.85) | 73.03 (-0.28) |
| | ResNet-50 | 76.12 | 64.16 (-11.96) | 73.59 (-2.53) | 75.71 (-0.41) | 76.04 (-0.08) |
| | ResNet-101 | 77.38 | 60.59 (-16.79) | 73.86 (-3.52) | 76.42 (-0.96) | 77.01 (-0.37) |

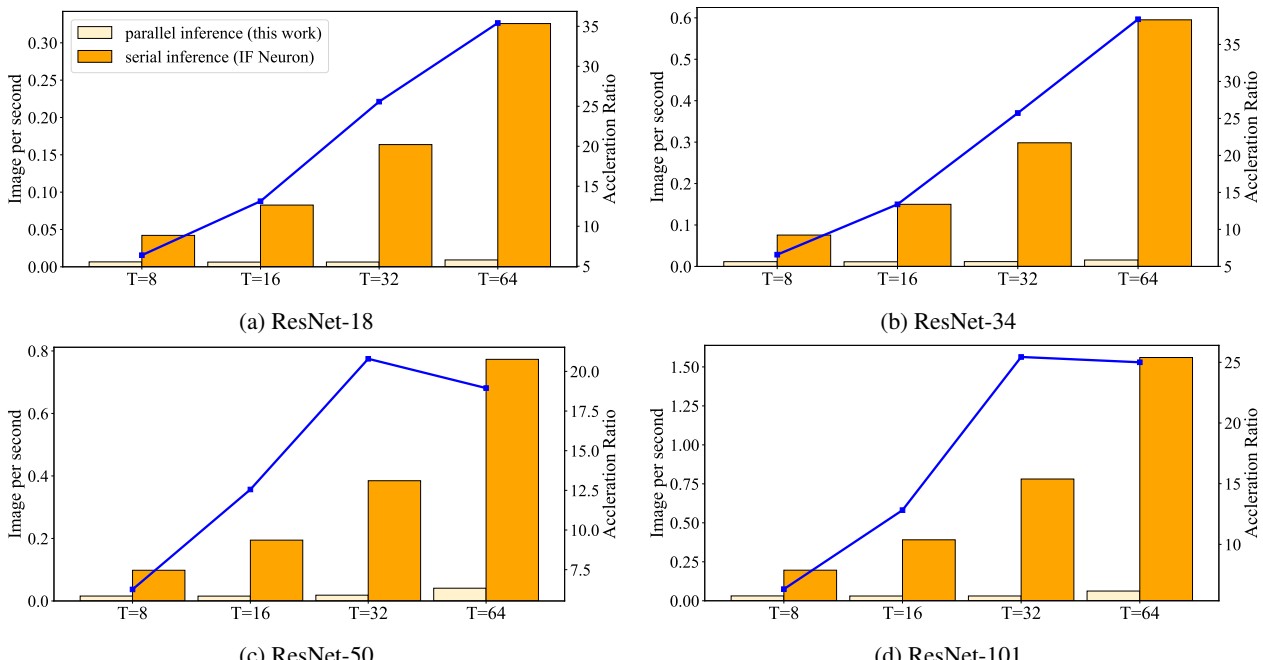

(a) ResNet-18 (b) ResNet-34

(c) ResNet-50 (d) ResNet-101

Figure 2: Comparison of parallel and serial inference speeds on ImageNet-1k dataset.

## 5.2. Performance Validation of Training-Free Parallel Conversion

We further investigate the parallel inference capability of our method under the condition of training-free conversion, as illustrated in Tab.4. One can find that our method can reduce the accuracy loss of conversion learning to $\leq 1\%$ within 32 time-steps and achieve better performance than previous schemes with only half of their inference latency. For example, we achieve accuracies of 66.26%(68.04%) on ResNet-18(34) within 16 steps, which exceeds the corresponding results of TBC within 32 steps by 15.61% and 9.01%, respectively. In addition, it is worth noting that even if the number of network layers increases to over 100 (*e.g.* ResNet-101), our training-free conversion framework can also rapidly squeeze the accuracy loss within the same time latency, preventing the conversion error from being exacerbated as the network becomes deeper.

## 5.3. Analysis of Parallel Inference Speed

As illustrated in Fig.2, we compare our parallel inference with the serial inference based on vanilla IF neuron. One can note that our scheme generally achieves $19 \sim 38\times$ acceleration ratio when $T \geq 32$, even for the very deep network backbone ($\geq 100$ layers). It is worth noting that the conversion error may be further amplified for complex network backbones or task scenarios, which leads to more severe time latency and performance degradation. Therefore, if we respectively consider the converted SNNs after adopting our method and traditional conversion framework at the same performance level, the actual advantages we achieve in terms of inference speed will become more remarkable. More experimental results can be found in Appendix.

# 6. Conclusion

In this paper, we propose a novel concept of parallel conversion and theoretically establish its mathematical equivalent relationship with the general activation function modules. Extensive experiments have validated that our scheme outperforms existing routes in SNN supervised learning in terms of inference performance and speed, which provides a brand-new approach for obtaining efficient SNN models.

# Acknowledgments

This work was supported by the National Natural Science Foundation of China (62422601, U24B20140, and 62088102), Beijing Municipal Science and Technology Program (Z241100004224004), Beijing Nova Program (20230484362, 20240484703), and National Key Laboratory for Multimedia Information Processing.

# Impact Statement

This paper presents work whose goal is to advance the field of SNN supervised learning. There are many potential societal consequences of our work, none which we feel must be specifically highlighted here.

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

# A. Appendix

## A.1. Proof of Theorem 4.1

**Theorem 4.1.** *For a $T$-steps parallel inference in the $l$-th layer, we use $\mathbf{b}^l$ to denote the corresponding shift term, here $\mathbf{b}^l \in \mathbb{R}^T$. When the pretrained ANN adopts QCFS function in Eq.(5), for the following cases, we will derive the optimal value of the shift term: $\mathbf{b}^l = \left[ \frac{\psi^l}{T} \cdots \frac{\psi^l}{T-x+1} \cdots \psi^l \right]^\top$.*

*(i) If $T = \tilde{T}$, then we have $\mathbf{r}^{l,T} = \mathbf{r}^{l,\tilde{T}}_{\mathrm{QCFS}}$.*

*(ii) If $T \neq \tilde{T}$ and $\psi^l = \theta^l/2$, then we have $\mathbb{E}\left( \mathbf{r}^{l,T} - \mathbf{r}^{l,\tilde{T}}_{\mathrm{QCFS}} \right) = \mathbf{0}$.*

*Proof.* (i) For $\mathbf{I}^l = \mathbf{W}^l \mathbf{r}^{(l-1),T} T \in \left[ k\theta^l - \psi^l, (k+1)\theta^l - \psi^l \right), \forall k \in [1, T]$, from Eq.(5) we can derive that $\mathbf{r}^{l,T}_{\mathrm{QCFS}} = k\theta^l/T$.

When we consider $\mathbf{s}^l = \left( \mathbf{\Lambda}^l_{\mathrm{PC}} \mathbf{I}^l + \mathbf{b}^l \geq \theta^l \right)$, we will have:

$$
\mathbf{s}^l = \left( \begin{bmatrix} \frac{1}{T} & \frac{1}{T} & \cdots & \frac{1}{T} \\ \frac{1}{T-1} & \frac{1}{T-1} & \cdots & \frac{1}{T-1} \\ \vdots & \vdots & \ddots & \vdots \\ 1 & 1 & \cdots & 1 \end{bmatrix} \mathbf{I}^l + \begin{bmatrix} \frac{\psi^l}{T} \\ \frac{\psi^l}{T-1} \\ \vdots \\ \psi^l \end{bmatrix} \geq \theta^l \right)
$$
$$
= \left( \begin{bmatrix} 1 & \cdots & \frac{T}{T-x+1} & \cdots & T \end{bmatrix}^\top \odot \mathbf{W}^l \mathbf{r}^{(l-1),T} + \begin{bmatrix} \frac{\psi^l}{T} & \cdots & \frac{\psi^l}{T-x+1} & \cdots & \psi^l \end{bmatrix}^\top \geq \theta^l \right). \tag{S1}
$$

Combining with $\mathbf{W}^l \mathbf{r}^{(l-1),T} \in \left[ \frac{k\theta^l - \psi^l}{T}, \frac{(k+1)\theta^l - \psi^l}{T} \right)$, we further have:

$$
\begin{bmatrix} \frac{k\theta^l}{T} & \cdots & \frac{k\theta^l}{T-x+1} & \cdots & k\theta^l \end{bmatrix}^\top \leq \mathbf{\Lambda}^l_{\mathrm{PC}} \mathbf{I}^l + \mathbf{b}^l < \begin{bmatrix} \frac{(k+1)\theta^l}{T} & \cdots & \frac{(k+1)\theta^l}{T-x+1} & \cdots & (k+1)\theta^l \end{bmatrix}^\top
$$
$$
\mathbf{s}^l = \left( \mathbf{\Lambda}^l_{\mathrm{PC}} \mathbf{I}^l + \mathbf{b}^l \geq \theta^l \right) \Rightarrow \mathbf{s}^l = \underbrace{[0 \; \cdots \; 1 \; \cdots \; 1]^\top}_{\text{firing } k \text{ spikes}} \tag{S2}
$$

Finally, we will derive $\mathbf{r}^{l,T} = \mathbf{r}^{l,T}_{\mathrm{QCFS}} = k\theta^l/T$.

(ii) Since QCFS function has the property of $\mathbb{E}\left( \mathbf{r}^{l,T}_{\mathrm{QCFS}} - \mathbf{r}^{l,\tilde{T}}_{\mathrm{QCFS}} \right) = \mathbf{0}$ when $\psi^l = \theta^l/2$, as mentioned in (Bu et al., 2022), combining with the conclusion of (i), we can have:

$$
\mathbb{E}\left( \mathbf{r}^{l,T} - \mathbf{r}^{l,\tilde{T}}_{\mathrm{QCFS}} \right) = \mathbb{E}\left( \mathbf{r}^{l,T} - \mathbf{r}^{l,T}_{\mathrm{QCFS}} \right) + \mathbb{E}\left( \mathbf{r}^{l,T}_{\mathrm{QCFS}} - \mathbf{r}^{l,\tilde{T}}_{\mathrm{QCFS}} \right) = \mathbf{0} + \mathbf{0} = \mathbf{0}. \tag{S3}
$$

Among them, $\mathbf{r}^{l,T} \to \mathbf{r}^{l,T}_{\mathrm{QCFS}}$ maintains lossless conversion under any precondition, while $\mathbf{r}^{l,T}_{\mathrm{QCFS}} \to \mathbf{r}^{l,\tilde{T}}_{\mathrm{QCFS}}$ needs to satisfy the assumption mentioned in (Bu et al., 2022) that the input current follows a uniform distribution within multiple sub-intervals. $\square$

## A.2. Detailed Experimental Configuration

For pretrained QCFS ANN models, we use SGD optimizer (Bottou, 2012), the optimization strategy of Cosine Annealing (Loshchilov & Hutter, 2017) and data augmentation techniques (DeVries & Taylor, 2017; Cubuk et al., 2019), the corresponding hyper-parameter settings are: $\mathtt{lr} = 0.1, \mathtt{wd} = 5 \times 10^{-4}$ for CIFAR-10, $\mathtt{lr} = 0.02, \mathtt{wd} = 5 \times 10^{-4}$ for CIFAR-100 and $\mathtt{lr} = 0.1, \mathtt{wd} = 1 \times 10^{-4}$ for ImageNet-1k. The specific network structure is consistent with (Bu et al., 2022). Regarding the error calibration technique, we utilize the training dataset as the calibration data to iterate for 1 epoch. The learning momentum $\alpha$ mentioned in Algorithm 1 is set to 0.99.

For ReLU ResNet family in Tab.4, we replace all ReLU modules except for Stem with ClipReLU, DA-QCFS, and parallel spiking neurons in sequence. The inference speeds in Fig.2 and Fig.S1 are measured on a single NVIDIA RTX 4090 GPU.

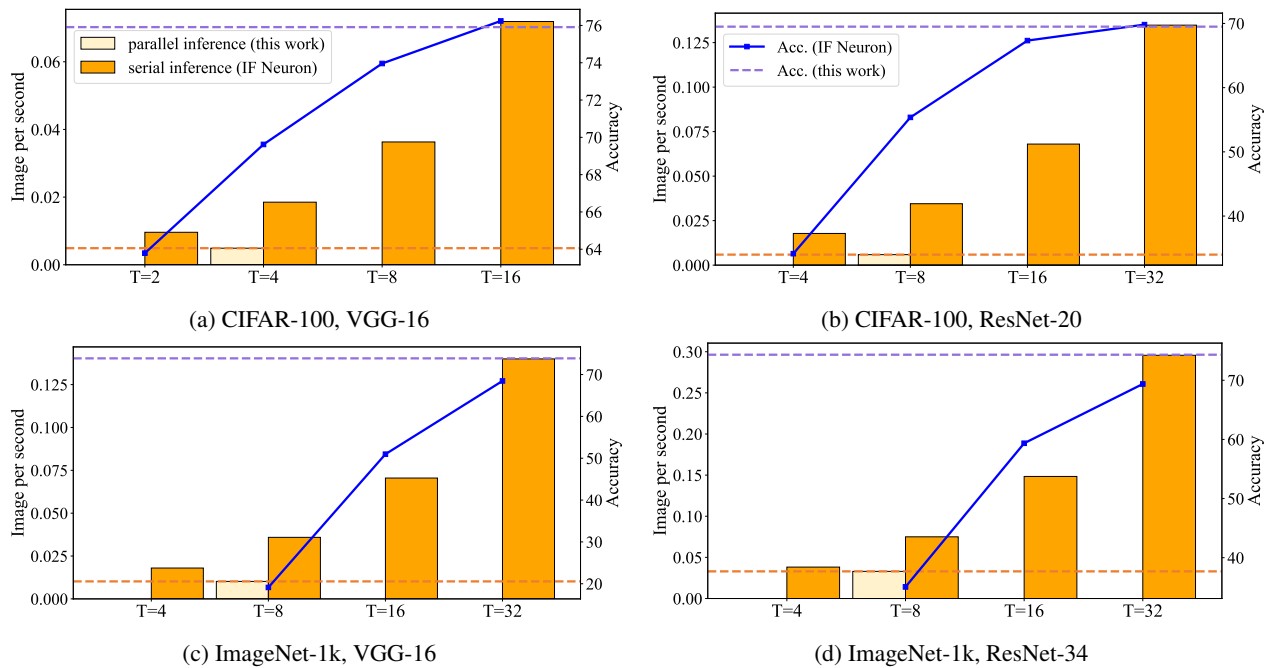

Figure S1: Comparison of parallel/serial inference speeds and performance on QCFS ANN models.

Among them, the inference speed of IF neuron is calculated on a subset of the test dataset (1000 images). In addition, experimental results reported in Tab.4, Fig.2 and Fig.S1 utilize $O(T)$ acceleration optimization in the charging phase.

## A.3. Comprehensive Analysis based on Inference Speed and Performance

As shown in Fig.S1, we make a comprehensive comparison between parallel and serial inference in terms of speed and learning accuracy. The serial inference performance under the QCFS conversion framework utilizes the accuracies reported in (Bu et al., 2022). For relatively simple cases (*i.e.* CIFAR-100 dataset), IF neuron requires approximately $4\times$ time latency of parallel conversion to reach the comparable accuracy, which makes our method achieve $15 \sim 23\times$ actual acceleration ratio.

For more complex scenarios (*i.e.* ImageNet-1k dataset), the ratio of time latency between IF neuron and parallel spiking neuron at the same performance level will further increase. Considering that our scheme can ensure lossless conversion within a specific time latency, this may lead to greater potential for parallel conversion in more challenging cases.

