# OpenReview forum: "Faster and Stronger: When ANN-SNN Conversion Meets Parallel Spiking Calculation"
_ICML.cc/2025/Conference — ICML 2025 poster_

### Official Review · Reviewer_mMJc · 2025-03-13

**Overall Recommendation:** 4

**Summary:**

This paper innovatively combines parallel spiking calculation with ANN-SNN Conversion to propose a high-performance and ultra-low-latency parallel conversion framework, which can also be applied in more general conversion scenarios (e.g. ReLU, QCFS with different quantization level). Experimental results have demonstrated that the superiority of the proposed method in terms of inference speed and performance.

**Claims And Evidence:**

The theoretical claim (e.g. Theorem 4.1) made in the submission is supported by convincing experimental validation.

**Essential References Not Discussed:**

At present, no previous work has been found to be omitted or improperly cited in this paper.

**Experimental Designs Or Analyses:**

According to Table 3-4 and Figure 2, compared to traditional conversion methods, parallel conversion achieves more superior performance within the same time latency.

**Methods And Evaluation Criteria:**

The derivation of the step-wise optimal shift term and the layer-wise error calibration based on DA-QCFS function ensure the fidelity of the conversion process.

**Other Comments Or Suggestions:**

The parallel conversion framework explores the performance upper-bound of parallel spiking computing (both precision and speed) from a new perspective.

**Other Strengths And Weaknesses:**

Strengths:
This paper explores the parallel spiking computation model from the perspective of conversion learning, establishing an equivalent mathematical mapping relationship between each time-step and the corresponding cumulative spike firing rate, which can also be considered as revealing the performance upper-bound of the parallel spiking model from another perspective. In addition, the authors further analyze the optimal value of the shift term and recognize the potential layer-by-layer distribution problem of the spike sequence, thus extending the proposed method to a more general conversion framework.
Weaknesses:
In Section 4.3, the authors mention that the computational cost of parallel conversion can be further optimized. I suggest that the authors can compare the specific computational overhead of vanilla IF model and parallel model in more detail from the perspective of operands (ADD, MUL), so as to make the parallel conversion scheme more convincing.

**Questions For Authors:**

SNNC-LP(AP) has also utilized an error calibration scheme before. What are the differences between the DA-QCFS based calibration method proposed in this work and SNNC-LP(AP)?

Li,Y., Deng,S., Dong,X., Gong,R., and Gu,S. A free lunch from ANN: Towards efficient, accurate spiking neural networks calibration. In International Conference on Machine Learning, 2021.

**Relation To Broader Scientific Literature:**

This paper provides a new perspective for further exploring the supervised learning schemes of SNNs (ANN-SNN Conversion, STBP Training, etc).

**Theoretical Claims:**

I have checked the correctness of the proof for theoretical claims in this paper.

---

> ### Author Rebuttal · Authors · 2025-03-30
>
> ## To Reviewer mMJc
>
> We are pleased that you recognize the relevant content of this work in terms of theoretical claims and experimental validation, as well as pointing out that our method provides a new perspective for SNN supervised learning. We will elaborate on your questions and comments.
>
> > The authors can compare the specific computational overhead of vanilla IF model and parallel model in more detail from the perspective of operands (ADD, MUL)
>
> **A1:** Thanks for the question. For the vanilla IF model, the inter-layer computation (assuming the synaptic layer is a fully-connected layer $[C,C]$) involves $O(T_\text{IF} NC^2)$ multiplication/addition operations, here $N,T_\text{IF}$ respectively denote the total number of tokens and time-steps. The intra-layer calculation involves charging, firing and resetting, with a total of approximately $2T_\text{IF}$ addition and $T_\text{IF}$ comparison operands for each neuron.
>
> For the parallel spiking model, the inter-layer computation is consistent with that of the IF model, while the intra-layer calculation can be combined with various optimization techniques proposed in Section 4.3. Specifically, when we convert $\mathbf{\Lambda}^l_\text{PC} \mathbf{I}^l$ to $[\frac{1}{T_\text{PC}},...,\frac{1}{T_\text{PC}-x+1},...,1]\odot\sum_t\mathbf{W}^l\mathbf{s}^{(l-1)}$, only $T_\text{PC}$ addition operands are involved, where $[\frac{1}{T_\text{PC}},...,\frac{1}{T_\text{PC}-x+1},...,1]$ can be further fused with $\theta^{l,t}$ at each time-step. This step involves $T_\text{PC}$ comparison operands. However, due to the existence of sorting property, the comparison operand can actually be further reduced to $O(\log T_\text{PC})$. Overall, using parallel conversion framework can save approximately $2T_\text{IF}-T_\text{PC}$ addition and $T_\text{IF}-O(\log T_\text{PC})$ comparison operands for each neuron layer, $O(T_\text{IF} NC^2)-O(T_\text{PC} NC^2)$ multiplication/addition operations for each synaptic layer, as well as achieving more superior inference performance and speed. Generally speaking, when traditional ANN-SNN Conversion reaches the same level of performance as Parallel Conversion, one can find that $T_\text{IF}\gg T_\text{PC}$.
>
> > What are the differences between the DA-QCFS based calibration method proposed in this work and SNNC-LP(AP)?
>
> **A2:** Thanks for the comment. SNNC-LP(AP) is committed to extremely low-cost error calibration from pre-trained ANNs to converted SNNs, where the average spike firing rate used for calibration is based on the assumption of uniform input current distribution. Due to SNNC-LP(AP) still using the vanilla IF model in the inference stage, there is still a gap between the estimated firing rate after calibration and the actual firing rate layer by layer.
> In comparison, the motivation of our calibration method is to achieve efficient inference of SNN under any time latency. Since parallel spiking calculation can satisfy the assumption of input current distribution, the process of replacing the rectified DA-QCFS module with parallel spiking neurons is lossless. The goal of calibration is to regulate the distribution of input current within a specified number of time-steps, so that the estimated firing rate of each layer is aligned with the highest learning accuracy as much as possible, thereby preparing for the parallel conversion process in advance.

---

> > ### Comment · Reviewer_mMJc · 2025-04-02
> >
> > Thanks for authors rebuttal. My concerns  have been addressed.

---

### Official Review · Reviewer_nBwe · 2025-03-16

**Overall Recommendation:** 4

**Summary:**

This work introduces a novel parallel conversion learning framework that establishes a mathematical mapping between each time step of parallel spiking neurons and the cumulative spike firing rate. The lossless and sorting properties of the conversion process are theoretically validated, and the optimal shifting distance for each step is identified. Additionally, it integrates distribution-aware error calibration to further improve accuracy.  Moreover,  the proposed framework achieves a top-1 accuracy of 72.90% on ImageNet-1k using ResNet-34 within only 4 time-steps, showcasing significant performance improvements in both conventional and training-free conversion scenarios.

**Claims And Evidence:**

Yes

**Essential References Not Discussed:**

[1] Scaling spike-driven transformer with efficient spike firing approximation training. IEEE T-PAMI 2025.

**Experimental Designs Or Analyses:**

Yes

**Methods And Evaluation Criteria:**

Yes

**Other Comments Or Suggestions:**

1. Please discuss in detail the difference between the conversion method in this paper and I-LIF [1], and whether the spike trains in this paper can be adjusted at will.

2. The proposed parallel transformation is limited to the convolutional architecture, and it remains an open question whether it can be applied to the Transformer architecture.





[1] Scaling spike-driven transformer with efficient spike firing approximation training. IEEE T-PAMI 2025.

**Other Strengths And Weaknesses:**

Strengths :

1. This paper proposes a low-time-step train-free ANN2SNN method to reduce the SNN training burden and inference cost.

2. Well-written and organized.

3. This work establishes a mathematical mapping between parallel spiking neurons and cumulative spike firing rates, theoretically validates the lossless and sorting properties of the conversion process, and derives the optimal shifting distance for each step.

Weaknesses:

1. The proposed parallel transformation is limited to the convolutional architecture, and it remains an open question whether it can be applied to the Transformer architecture.

2. The proposed parallel transformation can only convert fixed ANN architectures, and the characteristics of SNNs such as spike-driven may not be guaranteed during conversion.

**Questions For Authors:**

1. Please discuss in detail the difference between the conversion method in this paper and I-LIF [1], and whether the spike trains in this paper can be adjusted at will.

2. The proposed parallel transformation is limited to the convolutional architecture, and it remains an open question whether it can be applied to the Transformer architecture.





[1] Scaling spike-driven transformer with efficient spike firing approximation training. IEEE T-PAMI 2025.

**Relation To Broader Scientific Literature:**

N/A

**Theoretical Claims:**

Yes

---

> ### Author Rebuttal · Authors · 2025-03-30
>
> ## To Reviewer nBwe
>
> We are delighted that you think that our method is well written and organized, as well as being validated in both theoretical and experimental dimensions. We will discuss your questions in detail in the following content.
>
> > The difference between the conversion method in this paper and I-LIF.
>
> **A1:** Thanks for the question. From the perspective of conversion error, I-LIF transmits integer spikes during the training stage and can switch to vanilla LIF neurons during the inference stage. However, this can only ensure that I-LIF satisfies the input current assumption consistent with the QCFS function in the sub-slices composed of every $D$ time-steps ($D$ denotes the maximum integer spike value), and cannot fully achieve lossless conversion. In addition, I-LIF is commonly used in the field of STBP training. In comparison, parallel conversion is a learning framework specifically designed for ANN-SNN Conversion and has the property of lossless conversion.
>
> From the perspective of calculation mechanism, I-LIF, as an enhanced version of the vanilla LIF model, still consists of three processes: charging, firing and resetting. The overall computing process is serial and has a temporal direction. In comparison, our scheme adopts parallel spiking calculation, saving the processes of charging and resetting, and enabling more efficient SNN inference.
>
> >  Whether the spike trains in this paper can be adjusted at will?
>
> **A2:** For ANN-SNN Conversion, the key to learning lies in the average spike firing rate. As shown in Eq.8, our $\Lambda_\text{PC}^l$ is formed by the fusion of $\Lambda_\text{PRE}^l$ and $\Lambda_\text{POST}^l$, thus possessing the ability to homogenize the distribution of input current. In other words, theoretically, arbitrarily shuffling the order of the output spike sequence $\mathbf{s}^{(l-1)}$ will not affect the prediction of the average spike firing rate $\mathbf{r}^{l,T}$ in the next layer.
>
> From the perspective of adjusting the length of the spike sequence and the number of firing spikes, by combining the calibration scheme discussed in Sec 4.2 and Eq.9, we can arbitrarily adjust the length of the spike sequence and ensure that the converted SNN model has advanced performance within any time period.
>
> > It remains an open question whether the proposed parallel transformation can be applied to the Transformer architecture.
>
> **A3:** Thanks for the comment. Assuming that the input currents through $\mathbf{Q}^l,\mathbf{K}^l,\mathbf{V}^l$ are respectively $\mathbf{I}_Q^l,\mathbf{I}_K^l,\mathbf{I}_V^l\in\mathbb{R}^T$. If the input current is directly passed through the corresponding parallel spiking neurons $\text{SN}(\cdot)$ and the attention score is calculated, it will introduce potential computational complexity of $O(T^2)$. Therefore, one viable solution is to pre-calculate the average input current $\mathbf{I}_K^{l,\text{avg}},\mathbf{I}_V^{l,\text{avg}}\in\mathbb{R}$, then complete the calculation for $\left(\text{SN}(\mathbf{I}_Q^l){\mathbf{I}_K^{l,\text{avg}}}^{\top}\right)\mathbf{I}_V^{l,\text{avg}}$.
>
> At this point, one can note that the computational complexity of each step is at the level of $O(T)$ and always maintains $\mathbf{r}^{l,T}=\sum_t\mathbf{s}^{l,t}$. When $\mathbf{r}^{(l-1),T}=\sum_t\mathbf{s}^{(l-1),t}$, $\mathbf{I}^{l,x}=\Lambda_\text{PRE}^l\mathbf{W}^l\mathbf{s}^{(l-1)}=\mathbf{W}^l\mathbf{r}^{(l-1),T}, \forall x\in[1,T]$ holds, ensuring that the input current entering parallel neurons satisfies the distribution assumption and guarantees the precision of predicting the average spike firing rate layer by layer during the conversion process.

---

> > ### Comment · Reviewer_nBwe · 2025-04-02
> >
> > I appreciate your response and extra experiments. Most of the concerns have been addressed.

---

### Official Review · Reviewer_HMDR · 2025-03-17

**Overall Recommendation:** 4

**Summary:**

This paper propose a parallel ANN-SNN conversion framework. The author firstly categorizes and summarizes various conversion paradigms in the field of ANN-SNN conversion learning, then proposes an efficient conversion method based on parallel spiking computing, which relate each time-step to the cumulative spike firing rate. Experimental results show that the proposed methods can achieve SOTA performance on several benchmark datasets.

**Claims And Evidence:**

YES

**Essential References Not Discussed:**

nO

**Experimental Designs Or Analyses:**

The experimental comparison with previous SOTA methods has been checked.

**Methods And Evaluation Criteria:**

yES

**Other Comments Or Suggestions:**

For more comments or suggestions, please refer to the “Other Strengths And Weaknesses” section.

**Other Strengths And Weaknesses:**

Strengths
1.The theoretical analysis of the lossless property is solid.
2.The authors point out the non-uniform problem for both $[\mathbf{s}^{l,1},…,\mathbf{s}^{l,T}]$ and $\mathbf{r}^{l,T}, \mathbf{r}^{l,\tilde{T}}, T \neq \tilde{T}$, which is the foundation to establish theuniversal parallel conversion framework .
3.Compared to previous works, the performance advantage of parallel conversion is remarkable.

Weaknesses
1.The authors mainly focus on the classification performance of SNNs. Can this method be generalized to other tasks?
2.The author has verified the effectiveness of Parallel Conversion on CNN backbone. However, it lacks the test on Transformer-based SNNs. I suggest to add the analysis to make the contribution of this work more comprehensive.
3.The proposed method needs to train an ANN with quantization activation function, which brings training costs.

**Questions For Authors:**

As shown in Tab.1, the author further divides the concept of conversion learning into ANN-SNN Conversion (ReLU or QCFS), Conversion Rectification and Parallel Conversion. I am curious about the specific differences between Conversion Rectification and Parallel Conversion? For example, from the perspectives of performance and calculation overhead, etc.

**Relation To Broader Scientific Literature:**

The academic theme of this work is related to the ANN-SNN Conversion learning with high efficiency, the author respectively considers solutions for the non-uniform problem of the spike sequence and the simulated average firing rate.

**Theoretical Claims:**

The theoretical claims and proofs have been validated.

---

> ### Author Rebuttal · Authors · 2025-03-30
>
> ## To Reviewer HMDR
>
> We would like to thank for your acknowledgement about our approach in terms of theoretical analysis and performance advantages, we will provide further answers and clarifications for your questions and concerns.
>
> > Can this method be generalized to other tasks?
>
> Thanks for this comment. To validate the generalization ability of our method on visual tasks, we further conduct experimental verification on semantic segmentation tasks. We attempt training-free parallel conversion based on DA-QCFS for Pascal VOC dataset [1] and ResNet50-FCN/DeepLabv3 structure [2, 3], as shown in Tab. R1. Experimental results indicate that our scheme can also achieve approximately lossless parallel conversion in segmentation tasks.
>
> **Table R1:** Experimental results of training-free conversion on Pascal VOC dataset.
>
> |       Arch.        | Metric (%) |  ANN  |  T=8  | T=16  | T=32  |
> | :----------------: | :--------: | :---: | :---: | :---: | :---: |
> |    ResNet50-FCN    | Pixel Acc. | 88.76 | 86.99 | 88.12 | 88.60 |
> |    ResNet50-FCN    |    mIOU    | 51.93 | 46.71 | 49.70 | 51.25 |
> | ResNet50-DeepLabv3 | Pixel Acc. | 91.46 | 89.87 | 91.21 | 91.38 |
> | ResNet50-DeepLabv3 |    mIOU    | 63.03 | 58.91 | 62.48 | 62.72 |
>
>
>
> > The analysis of Parallel Conversion for Transformer-based SNNs?
>
> **A2:** Thanks for the comment. The most critical difference between Transformer and CNN is the matrix multiplication between multi-branch inputs (e.g. $\mathbf{Q}^l{\mathbf{K}^l}^{\top}, \text{Attn}^l\mathbf{V}^l$). If we directly insert QCFS functions after $\mathbf{Q}^l$ and $\mathbf{K}^l$ weight layers in the pre-training stage of ANN, and then replace QCFS modules with parallel spiking neurons $\text{SN}(\cdot)$ in the SNN inference stage, this may introduce $O(T^2)$ computational complexity within the self-attention modules. Therefore, we can consider pre-calculating the average input current for one of the branches (e.g. $\mathbf{I}^{l,\text{avg}}_Q$) and then calculating the attention score $\mathbf{I}^{l,\text{avg}}_Q \text{SN}({\mathbf{I}^l_K}^{\top})$ to maintain the computational complexity at $O(T)$. It is worth noting that at this point, parallel conversion will still maintain its unique lossless and sorting properties on Transformer-based SNNs.
>
> > The proposed method needs to train an ANN with quantization activation function, which brings training costs.
>
> **A3:** For parallel conversion based on QCFS ANN, we usually need to complete the pre-training process of the quantized ANN model. However, our method also validate its effectiveness in training-free conversion cases. We can directly obtain the corresponding converted SNN model through utilizing quantization, error calibration, and parallel neuron replacement on an open-source and training-free ANN checkpoint.
>
> > The specific differences between Conversion Rectification and Parallel Conversion from the perspectives of performance and calculation overhead?
>
> **A4:** Thanks for the question.  The concept of Conversion Rectification usually refers to the relevant schemes for secondary optimization of the converted SNN model in the inference stage, aiming to reduce the representation gap between pre-trained ANNs and converted SNNs. The previous related works were generally based on IF neuron and its variants, and most of the works cannot guarantee the complete elimination of conversion errors in theory. Compared to these methods, our parallel conversion has faster inference speed and the property of lossless conversion . Additionally, due to its small number of time-steps, it can always keep the calculation overhead within a reasonable range.
>
> [1] The PASCAL Visual Object Classes (VOC) Challenge, IJCV, 2010.
>
> [2] Fully convolutional networks for semantic segmentation. CVPR, 2015.
>
> [3] Rethinking Atrous Convolution for Semantic Image Segmentation, 2017.

---

### Official Review · Reviewer_72NY · 2025-03-19

**Overall Recommendation:** 3

**Summary:**

This work presents a novel route for SNN supervised learning by jointly adopting ANN-SNN Conversion and parallel calculation. The main contributions of the paper include the proof of optimal shifting distance, further promotion of parallel conversion framework based on QCFS, and experimental demonstration on various benchmarks.

**Claims And Evidence:**

Yes

**Essential References Not Discussed:**

No

**Experimental Designs Or Analyses:**

I have checked the experimental results about the Accuracy and Acceleration Ratio of this method.

**Methods And Evaluation Criteria:**

Yes

**Other Comments Or Suggestions:**

The authors' proof of the theorem in Appendix is somewhat concise, and more additional explanation can be introduced to enhance its readability.

**Other Strengths And Weaknesses:**

Strengths
1. The narrative logic of this work is clear. Experiments have shown that parallel conversion has significant advantages in accuracy and acceleration ratio compared to traditional conversion methods.
Weaknesses：
1. The discussion on threshold recording and error calibration techniques is not sufficient. For example, as shown in Table 2, the authors need to further explain why some cases require the use of the above two techniques, while others do not?
2. In Section 4.3, the author needs to clarify more clearly which calculation step does the “sorting property” save in terms of computational cost?

**Questions For Authors:**

1: In Section 5.3 (Figure 2) and Appendix A.2 (Figure S1), how are the corresponding throughput rates calculated for serial inference based on IF model and parallel inference based on parallel spiking model? Please clarify it.
2: For ResNet-50 and ResNet-101 (Figure 2.c-2.d), it seems that their corresponding Acceleration Ratio do not further increase like other cases when the number of time-steps is large ($T\geq 32$). What is the specific reason for this? Does it means the proposed method work only for low-latency SNNs?

**Relation To Broader Scientific Literature:**

This work investigates a new conversion method that combines the performance advantages of ANN-SNN conversion with the low-latency advantage of STBP Training in the inference stage. It is a new learning route in SNN field.

**Theoretical Claims:**

I have checked the relevant proof

---

> ### Author Rebuttal · Authors · 2025-03-30
>
> ## To Reviewer 72NY
>
> We sincerely appreciate your recognition for the novelty and experimental effectiveness of this work. We will strive to address your concerns in detail in the following section:
>
> > The discussion on threshold recording and error calibration techniques is not sufficient.
>
> **A1:** Thanks for the question. In this work, threshold recording is used to confirm the initial maximum threshold and effectively utilize the distribution of input current, as QCFS explicitly provides learnable thresholds during the pre-training stage of ANN, threshold recording is only used in training-free conversion based cases. The motivation of error calibration lies in the fact that QCFS can only eliminate conversion errors ($T\neq\tilde{T}$) when the input current follows a uniform distribution, as shown in Theorem 4.1(ii). However, since the assumed condition may not be fully satisfied in practical situations, additional error calibration can further enhance the performance of converted SNNs under low time latency (where ClipReLU can also be considered as a case of $\tilde{T}\gg T$).
>
> > Which calculation step does the “sorting property” save in terms of computational cost?
>
> **A2:** The sorting property actually effectively saves multiplication operations in $\Lambda_\text{PC}^l\mathbf{I}^l+\mathbf{b}^l$ and comparison operations between $\Lambda_\text{PC}^l\mathbf{I}^l+\mathbf{b}^l$ and $\theta^l$. Specifically, when we utilize the sorting property, it actually only involves calculation related to $\Lambda_\text{PC}^{l,\text{idx}}\mathbf{I}^l+\mathbf{b}^{l,\text{idx}}\geq\theta^{l,\text{idx}}$, where $\text{idx},\text{len(idx)}=O(\log T)$ is the index time-step set selected by the sorting property.
>
> > How are the corresponding throughput rates calculated?
>
> **A3:** We randomly select a subset of data (1000 images) to calculate the average inference time for the above two schemes. We choose scientific tools to calculate the precise time and the specific process has been submitted into the supplementary materials. For serial inference, we feed data into the SNN backbone at each time-step, while parallel inference packages consecutive $T$ time-steps into a batch and feeds it into the network all at once.
>
> > For ResNet-50 and ResNet-101, it seems that their Acceleration Ratio do not further increase like other cases?
>
> **A4:** Thanks for this insightful comment. Due to the fact that $T$ time-steps will be passed into the corresponding network backbone at once during parallel inference, for experimental cases with large model parameter quantity or time-steps, the required hardware memory is large and the utilization rate tends to be saturated. Therefore, under a environment with limited hardware support, the corresponding acceleration ratio may not further increase. However, even so, the advantage of parallel conversion is very obvious when $T\geq 32$ (e.g. the corresponding acceleration ratio exceeds $17.5\times$ for Fig 2.c-d). In addition, due to the theoretically lossless property of our method, in practical scenarios, we usually only need a small number of time-steps to achieve the same level of performance as the pretrained ANN for the converted SNN.
>
> > The authors' proof of the theorem in Appendix is somewhat concise.
>
> **A5:** Thanks for the suggestion. We will further improve the relevant proof process in the final submission.

---

### Decision · Program_Chairs · 2025-05-01

**Decision:**

Accept (poster)

**Comment:**

This paper presents a novel and technically rigorous framework for parallel ANN-SNN conversion, combining parallel spiking computation with distribution-aware calibration. The key contributions include: (1) a mathematically grounded mapping between time-steps and cumulative spike firing rates; (2) theoretical validation of the lossless and sorting properties of the conversion process; (3) integration of an efficient error calibration technique to improve accuracy across various quantization levels and training regimes; and (4) strong empirical results on standard classification and segmentation benchmarks, with particular emphasis on ultra-low-latency inference.

The reviewers were generally positive, with three reviewers recommending acceptance (score: 4) and one reviewer leaning toward acceptance (score: 3). All acknowledged the strong theoretical contributions, clear writing, and empirical superiority of the proposed method compared to existing ANN-SNN conversion frameworks.

Strengths noted by reviewers include:

* Well-structured mathematical formulation and proof of conversion properties.

* Significant reduction in latency and computational cost, demonstrated across multiple architectures and datasets.

* Support for both training-free and training-based settings, with practical implementation details included in the supplementary material.

* Generalization beyond classification to semantic segmentation tasks, improving the perceived applicability of the method.

Concerns raised by reviewers were adequately addressed in the rebuttal:

* The authors provided detailed explanations regarding error calibration use cases, the impact of sorting properties, and implementation details of throughput measurements.

* Clarified the method’s applicability to Transformer-based SNNs and addressed differences with prior works like I-LIF and SNNC-LP(AP).

* Quantified computational savings (ADD/MUL/comparisons) compared to vanilla IF models, reinforcing the efficiency claims.

* Demonstrated the method’s generalization on segmentation tasks and discussed future directions for more diverse architectures.

Overall, the paper proposes a well-founded and practically impactful advancement in SNN training and inference. The rebuttal was thoughtful and resolved the reviewers’ concerns effectively, and the discussion reflects maturity in theoretical and empirical SNN research.